# Glycogen Storage Disease Phenotypes Accompanying the Perturbation of the Methionine Cycle in NDRG3-Deficient Mouse Livers

**DOI:** 10.3390/cells11091536

**Published:** 2022-05-04

**Authors:** Hyun Ahm Sohn, Dong Chul Lee, Anna Park, Minho Kang, Byoung-Ha Yoon, Chul-Ho Lee, Yong-Hoon Kim, Kyoung-Jin Oh, Cha Yeon Kim, Seong-Hwan Park, Han Koo, Hyoung-Chin Kim, Won Kee Yoon, Dae-Sik Lim, Daesoo Kim, Kyung Chan Park, Young Il Yeom

**Affiliations:** 1Department of Biological Sciences, Korea Advanced Institute of Science and Technology, Daejeon 34141, Korea; imcm1109@kaist.ac.kr (H.A.S.); daesiklim@kaist.ac.kr (D.-S.L.); daesoo@kaist.ac.kr (D.K.); 2Personalized Genomic Medicine Research Center, Korea Research Institute of Bioscience and Biotechnology (KRIBB), Daejeon 34141, Korea; dclee@kribb.re.kr (D.C.L.); mhkang@kribb.re.kr (M.K.); bluesky6711@gmail.com (C.Y.K.); shpark1@kribb.re.kr (S.-H.P.); koohan12@gmail.com (H.K.); 3Metabolic Regulation Research Center, Korea Research Institute of Bioscience and Biotechnology (KRIBB), Daejeon 34141, Korea; annapark@kribb.re.kr (A.P.); kjoh80@kribb.re.kr (K.-J.O.); 4Korea Bioinformation Center, Korea Research Institute of Bioscience and Biotechnology (KRIBB), Daejeon 34141, Korea; cogate@kribb.re.kr; 5Laboratory Animal Resource Center, Korea Research Institute of Bioscience and Biotechnology (KRIBB), Daejeon 34141, Korea; chullee@kribb.re.kr (C.-H.L.); yhoonkim@kribb.re.kr (Y.-H.K.); 6Department of Functional Genomics, KRIBB School of Bioscience, University of Science and Technology, Daejeon 34141, Korea; 7Laboratory Animal Resource Center, Korea Research Institute of Bioscience and Biotechnology, Cheongju 28116, Korea; hckim@kribb.re.kr (H.-C.K.); wkyoon@kribb.re.kr (W.K.Y.)

**Keywords:** NDRG3, glycogen storage disease, PYGL, methionine cycle, reprogramming, GNMT

## Abstract

N-Myc downstream regulated gene 3 (NDRG3) is a unique pro-tumorigenic member among NDRG family genes, mediating growth signals. Here, we investigated the pathophysiological roles of NDRG3 in relation to cell metabolism by disrupting its functions in liver. Mice with liver-specific KO of NDRG3 (Ndrg3 LKO) exhibited glycogen storage disease (GSD) phenotypes including excessive hepatic glycogen accumulation, hypoglycemia, elevated liver triglyceride content, and several signs of liver injury. They suffered from impaired hepatic glucose homeostasis, due to the suppression of fasting-associated glycogenolysis and gluconeogenesis. Consistently, the expression of glycogen phosphorylase (PYGL) and glucose-6-phosphate transporter (G6PT) was significantly down-regulated in an Ndrg3 LKO-dependent manner. Transcriptomic and metabolomic analyses revealed that NDRG3 depletion significantly perturbed the methionine cycle, redirecting its flux towards branch pathways to upregulate several metabolites known to have hepatoprotective functions. Mechanistically, Ndrg3 LKO-dependent downregulation of glycine N-methyltransferase in the methionine cycle and the resultant elevation of the S-adenosylmethionine level appears to play a critical role in the restructuring of the methionine metabolism, eventually leading to the manifestation of GSD phenotypes in Ndrg3 LKO mice. Our results indicate that NDRG3 is required for the homeostasis of liver cell metabolism upstream of the glucose–glycogen flux and methionine cycle and suggest therapeutic values for regulating NDRG3 in disorders with malfunctions in these pathways.

## 1. Introduction

N-Myc downstream regulated genes (NDRGs) are a highly conserved family of genes implicated in cell proliferation, differentiation, development, and stress responses [1]. They consist of four members (NDRG1-4) sharing an α/β-hydrolase fold region located within a conserved NDR domain, and thereby belong to the α/β-hydrolase fold superfamily. Since all of the NDRG members lacks the catalytic triad of a typical α/β-hydrolase fold due to amino acid substitutions, i.e., nucleophile-acid-histidine, they are considered enzymatically nonfunctional [1,2] but, instead, are thought to play nonenzymatic functions mainly involving protein–protein interactions [3].

Accumulating evidence indicates that NDRGs are intimately associated with tumorigenesis in diverse tissue types [1,4]. Notably, among four members of the family, NDRG1, 2, and 4 largely mediate tumor suppressive functions, while NDRG3 is associated with pro-tumorigenic functions. Thus, NDRG1, 2, and 4 are associated with good prognosis of cancer patients but show mostly downregulated expression in different cancers. On the other hand, NDRG3 expression is frequently upregulated in many cancer types, including prostate cancer, hepatocellular carcinoma, nonsmall cell lung cancer, breast cancer, and colorectal cancer, in association with aggressive cancer phenotypes and a poor prognosis [5,6,7,8,9,10].

Cell metabolism is a critical determinant of the malignant characteristics of a cancer [11], and potential roles of NDRGs in reprogramming cancer metabolism under hypoxia have been suggested [4]. In fact, existing studies on the role of different NDRGs in cellular metabolism seem to reasonably coincide with their characteristics in tumor biology. For example, the tumor suppressive NDRG members are implicated in the inhibition of anabolic metabolisms. In one aspect of the metabolic regulation, tumor suppressive NDRGs inhibit the expression of genes involved in glycolysis and glutaminolysis by suppressing HIF1- or Myc-mediated gene transcription [12,13] or by promoting GLUT1 protein degradation [14]. In addition, tumor suppressive NDRGs negatively regulate the activity of anabolic metabolism via their antagonistic effects on the PI3K-AKT-mTOR network, which is a major growth regulatory pathway promoting anabolic programs involving increased glycolytic flux and fatty acid synthesis [15,16,17].

In spite of the potentially unique position of NDRG3 in tumor biology among the NDRG members, its possible roles in the metabolism of cancer or normal cells were barely explored. NDRG3 promoted prostate cancer cell growth, along with the upregulation of angiogenic chemokine expression [5] and facilitated colorectal cancer metastasis by activating Src phosphorylation [10]. NDRG3 promoted tumor growth and angiogenesis in HCC via signaling through Raf-ERK axis [18], and enhanced HCC metastasis via Wnt/β-catenin signaling by promoting nuclear translocation of β-catenin [7]. Considering the pro-tumorigenic characteristics of NDRG3 action, it may be associated with anabolic functions of metabolism. In this study, in an effort to understand the pathophysiological roles of NDRG3 in relation to cell metabolism, we generated mice having their *Ndrg3* gene specifically disrupted in hepatocytes. We showed that NDRG3 depletion in the liver causes GSD phenotypes in mice due to the malfunctioning of the glycogen degradation pathway, and that NDRG3 deficiency-induced dysregulation of the methionine cycle is intimately associated with the manifestation of these pathologic phenotypes.

## 2. Materials and Methods

### 2.1. Animals

Mice were maintained in a specific pathogen-free facility of the Korea Research Institute of Bioscience and Biotechnology (KRIBB) following the instructions from the Institutional Animal Care and Use Committee (IACC). Experiment mice were maintained at 23 °C and 40–50% humidity under a 12 h light–dark cycle and allowed to freely access water and a normal chow diet (Envigo, Indianapolis, IN, USA). Male mice at the age of 4–12 weeks were used in each experiment unless otherwise specifically mentioned. For food restrictions, mice were fasted overnight (15 h) and/or refed for 3 or 6 h.

### 2.2. Generation of Liver-Specific Conditional Ndrg3 Knockout (Ndrg3 LKO) Mouse

Chimeric mice harboring the *Ndrg3* conditional knockout allele were purchased from the Knockout Mouse Project (KOMP) Repository and bred according to the manufacturer’s instructions. Briefly, the chimera mice were backcrossed with C57BL/6N mice for more than five generations for selection by coat color and genotyping, and a heterozygous *Ndrg3*^f/+^ offspring was obtained. The *Ndrg3*^f/+^ mouse was mated with a transgenic mouse expressing Flp recombinase to remove β-geo selection sites of embryonic cells. The resulting heterozygous offspring (*Ndrg3*^f/+^) were mated with each other and a homozygous floxed mouse (*Ndrg3*^f/f^) was generated and genotyped for germ line transmission. Subsequently, the homozygous floxed *Ndrg3* mouse (*Ndrg3*^f/f^) was bred with a transgenic mouse expressing Alb-Cre recombinase to establish a mouse harboring the liver-specific deletion allele. Finally, the heterozygous (*Ndrg3*^f/+^; Alb-Cre) and homozygous *Ndrg3*^f/f^ mice were mated, and Ndrg3 LKO were established. The wild-type littermates were used as the control.

### 2.3. Glucose Tolerance Test (GTT)

For the glucose tolerance test, mice were fasted overnight and then injected with D-glucose (Sigma-Aldrich, St. Louis, MO, USA, #G8270) at a dose of 2 g/kg body weight by intraperitoneal (i.p.) administration. Subsequently, blood glucose levels were evaluated with portable blood glucose meter (Accu-check, Roche, Basel, Switzerland) at 0, 15, 30, 45, 60, 90, and 120 min.

### 2.4. Isolation of Primary Hepatocytes and Examination of Insulin/Glucagon Signaling

Primary hepatocytes were isolated by two-step collagenase method as reported [19]. Briefly, primary hepatocytes were extracted from the liver of a C57BL/6 mouse at the age of 8-12 weeks. The mouse was anesthetized with avertin (250 mg/kg; Sigma-Aldrich, #T48402) with i.p. injection, and the liver was perfused with HBSS media (Ca++ free) containing 1 mM EGTA (Sigma-Aldrich, #0396) followed by a collagenase solution with HBSS media (Ca++) containing 0.05% collagenase type IV (Sigma-Aldrich, #5138) through the inferior vena cava, whereas it was drained by portal vein cutting. Subsequently, detached hepatocytes were filtered with a 70 um pore-size mesh, centrifuged at 50× *g* for 4 min twice and subjected to purification by 40% Percoll gradients (GE Heaklthcare, Uppsala, Sweden, #17-0891-01). Primary hepatocytes were used in subsequent experiments only when cell viability was above 90% as measured by the automated cell counter (Thermo Scientific, Waltham, MA, USA) after trypan blue staining. Primary hepatocytes were seeded with Dulbecco’s Modified Eagle Medium (DMEM) (Welgene, Gyeongsan, Korea, #LM001-05) or M199 medium (Thermo Fisher Scientific Inc., Grand Island, NY, USA, #11150-059) in Corning^®^ Primaria™ 6-well plates (Corning Inc., New York City, NY, USA, #353846) supplemented with 100 nM dexamethasone (Sigma-Aldrich, #D4902), 10% fetal bovine serum (Thermo Fisher Scientific Inc., #16000044), and 1% penicillin (Thermo Fisher Scientific Inc., #15140-122). Cells were washed twice with PBS and maintained with fresh media for 4 h after seeding in 6-well plates. For insulin stimulation, hepatocytes stabilized for 4 h after seeding in 6-well plates were then washed with PBS four times and incubated overnight with serum-deprived DMEM containing penicillin (1%). The cells were then treated with 100 nM insulin (Sigma-Aldrich, #I9278) for 0, 10, 30, and 60 min and lysed with RIPA buffer (NaCl 150 mM, KCl 100 mM, HEPES 20 mM, EDTA 10 mM, 1% Triton-X) after washing with PBS three times. For the glucose production experiment, primary hepatocytes isolated from 2-month-old wild and Ndrg3 LKO mice were plated and incubated with serum-free medium overnight. Before the assay, the hepatocytes were changed to glucose-free DMEM for 2 h followed by PBS washing. Subsequently, the hepatocytes were incubated with glucose production buffer (glucose-free DMEM (pH 7.4) without phenol red supplemented with 20 mM sodium lactate, 2 mM sodium pyruvate, 2 mM L-glutamine, and 15 mM HEPES) with or without 100 nM glucagon (Sigma-Aldrich, #G2044) and were kept in the same buffer for 4 h. The medium was collected and subject to measurement of the glucose concentration by a colorimetric glucose assay kit. The readouts were then normalized by the total protein content determined from whole-cell lysates, and the glucose production rate of glucagon-stimulated hepatocytes was presented as the fold change relative to vehicle-treated primary hepatocytes.

### 2.5. Liver Histology

Liver tissues were extracted from mice and immediately immersed in 10% neutral buffered formalin (BBC Biochemical Corp, Seattle, WA, USA) for 24 h, paraffin-embedded, sectioned in 4 um thickness using microtome, and mounted on the silane-coated slide. For hematoxylin and eosin staining, tissues were deparaffinized and hydrated by serial incubation in alcohol diluted serially from 100% to 50%. After rinsing, the tissue sections were incubated with hematoxylin followed by eosin and mounted in resin after dehydration by alcohol followed by xylene.

### 2.6. Periodic Acid Schiff (PAS) Staining

Glycogen in tissues or primary hepatocytes was detected using a PAS Stain Kit (Abcam, Cambridge, UK, #ab150680) according to the manufacturer’s instruction. Briefly, sectioned tissues were deparaffinized, hydrated with distilled water, and incubated with periodic acid solution for 5 min and rinsed with distilled water. In addition, tissues were incubated with Schiff’s reagent for 15 min and rinsed with running tap water and counterstained with hematoxylin (modified Mayer’s) for 1 min. Mounted tissues were visualized with an Olympus microscope.

### 2.7. Protein Analysis

Primary hepatocytes were lysed with RIPA buffer (Thermo Scientific, Rockford, lL, USA, #78510) containing protease inhibitor (Roche, #11836170001) and phosphatase inhibitor cocktail (Roche, #4906837001) for 30 min and centrifuged at 12,000 rpm at 4 °C for 10 min. Protein concentration from supernatant was assessed using Bredford solution (BioRad, Hercules, CA, USA, #5000006). Western blotting was executed. Briefly, protein (40–50 ug) was loaded in SDS-PAGE gel with 5× sample buffer, transferred onto a nitrocellulose membrane, incubated with primary antibody overnight, and then with horseradish peroxidase(HRP)-conjugated secondary antibody for 1 h at RT, and visualized with chemiluminescence detection (Pierce Biotechnology Inc., Rockford, IL, USA, #34577) using LAS-4000 image analyzer (Fujifilm Inc., Stanford, CT, USA). HSP90 was used as a loading control for each protein volume. Antibodies were purchased as follows: Antibodies against pS473-AKT(#4060), AKT (#9272), pS9-GSK-3β (#9323), GSK-3β (#9315), pS641-GS (#3891), GS (#3886), and HSP90 (#4874) were from Cell Signaling Technology (CST, Danvers, MA, USA). Antibodies against GNMT (#18790-1-AP), and PYGL(#15851-1-AP) were from Proteintech Group Inc. (Rosemont, IL, USA). Protein phosphatase 1 antibody (PP1, sc-7482) was from Santa Cruz Biotechnology (SCBT Inc., Dallas, Texas, USA). pS15-PYGL antibody (#S961A) was from Division of Signal Transduction Therapy (DSTT; University of Dundee, UK). NDRG3 polyclonal antibody was customized from a commercial facility (AB Frontier, Seoul, Korea). This antibody was designed to recognize the N-terminal portion of NDRG3 protein by immunizing rabbits with an NDRG3 peptide (LNDKNGTRNFQDFDC; amino acids 16–30).

### 2.8. Glycogen Measurement

Glycogen levels in liver tissues were measured using a Glycogen Assay Kit (Abnova, Taipei City, Taiwan, #KA0861) following the manufacturer’s instruction. Snap-frozen liver tissues (10 mg) were homogenized and diluted with 200 uL dH_2_O. The homogenates were boiled at 100 °C for 10 min, centrifuged at 12,000 rpm for 10 min to remove insoluble material, and then the supernatants were collected. The samples were hydrolyzed by incubation with hydrolysis enzyme mixture containing glucoamylase converting glycogen to glucose for 30 min at room temperature. Subsequently, the samples were incubated with a reaction mixture for oxidation reaction, and the resulting, colored product was detected for optical density (OD) at 570 nm.

### 2.9. Glucose Measurement

Glucose level was measured with a Glucose Colorimetric Assay Kit (BioVision, Milpitas, CA, USA, #K606-100). Glucose from liver tissue was oxidized and reacted with a dye to generate a colored product by incubation with an enzyme mix for 30 min. Color absorbance was measured with a microplate reader (OD 570 nm).

### 2.10. Glucose-6-Phosphate (G6P) Measurement

G6P level was determined by a Glucose-6-Phosphate Colorimetric Assay Kit (BioVision, #K657-100). Briefly, tissue samples were diluted and homogenized with ice-cold PBS, boiled at 100 °C for 10 min to inactivate enzymes, and centrifuged to remove insoluble materials. The supernatant containing G6P was incubated with the supplied specific dye probe and enzyme mixture for oxidization and conversion of colorless probe to colored product. OD was evaluated by a microplate reader at 450 nm.

### 2.11. Glucose-1-Phosphate (G1P) Measurement

A Glucose-1-Phosphate Colorimetric Assay (G1P) Kit (BioVision, #K697-100) was utilized for determining the G1P level. Briefly, tissue samples were diluted and homogenized with ice-cold G1P assay buffer, boiled at 100 °C for 10 min to inactivate enzymes hampering the assay reaction, and centrifuged to remove insoluble materials. The supernatant including G1P was incubated with the supplied enzyme mixture containing phosphoglucomutase and glucose 1,6-biphosphate for conversion to G6P, leading to the oxidation of G6P to generate NADH, converting the probe into colored products. OD was assessed by a microplate reader at 450 nm.

### 2.12. Glycogen Phosphorylase L (PYGL) Activity Measurement

PYGL activity was assayed using a Glycogen Phosphorylase Assay Kit (Abcam, ab273271). Briefly, tissue samples (50 mg) were diluted and homogenized with ice-cold assay buffer, incubated on ice for 15 min, and centrifuged to remove insoluble materials. The samples were incubated with glycogen and reaction mixture and measured at OD 450 nm to detect the conversion of glycogen to G1P.

### 2.13. Glycogen Synthase (GS) Activity Measurement

A Glycogen Synthase Microplate Assay Kit (Biorbyt Ltd, Cambridge, United Kingdom, orb707344) was used for detection and quantification of GS activity. Briefly, tissue samples (100 mg) were diluted and homogenized with 1 mL assay buffer on ice and centrifuged to remove insoluble materials. The samples were incubated with the enzyme mixture and calorimetric readout was assessed at 340 nm by determining NADH decomposition rate.

### 2.14. Real-Time RT-PCR

RNA for primary hepatocytes was prepared with an RNA extraction kit (Qiagen) following the manufacture’s description. RNA quality was checked in Nanodrop One (Thermo Scientific). Complementary DNA (cDNA) was synthesized using RNA (5 ug) with reverse transcriptase (Thermo Scientific). Real-time PCR was performed using PCR master mix (SYBR Green Applied Biosystem, Waltham, MA, USA, #4367659) and monitored on CFX connect real-time system (Bio-Rad, #1855201). Reaction conditions were as follows: 95 °C, 5 min, followed by 45 cycles of 95 °C for 10 s for denaturation, 60 °C for 30 s for annealing, and 60 °C for 30 s for elongation. Duplicate samples per mouse tissue were harvested and, for relative quantification, the experimented genes were normalized using *Gapdh* as the reference, and then calculated using the comparative Ct method. Primer sequences are listed in Appendix A, and all oligonucleotides were synthesized from Bioneer Inc. (Daejeon, Korea).

### 2.15. Metabolome Analysis

For the metabolome analysis, liver samples (5 mg per tissue) extracted from wild-type (*n* = 5) and Ndrg3 LKO (*n* = 5) mice at the age of 8–10 weeks were snap-frozen in liquid nitrogen and transferred to Human Metabolome Technologies, Inc. (HMT Inc., Yamagata, Japan). Samples were analyzed according to capillary electrophoresis-time-of-flight mass spectrometry (CE-TOFMS) in two modes for cationic and anionic metabolites. For sample preparation, the samples were mixed with 50% acetonitrile in water (*v*/*v*) containing internal control (10 µL), homogenized and supernatants were filtered with cut-off filter (5-KDa, Ultrafree-MC-PLHC, HMT). The filtrate was concentrated and diluted with 50 uL ultrapure water. Automatic integration software (MasterHands ver. 2.17.1.11, Keio University) was utilized to extract peaks detected in CE-TOFMS analysis and to obtain the peak information about m/z, migration time (MT), and peak area. On the basis of tolerance of ±10 ppm in m/z and ±0.5 min in MT, HMT’s standard and the Known-Unknown peak library provided information for assigned metabolites. A volcano plot was used to show the differential change of metabolome data in terms of statistical significance *versus* magnitude of change. As input data, fold-change and *p*-value of metabolome data were used. A volcano plot was created using house-made Rscipt. In a volcano plot, the most up- and down-regulated metabolites are towards the right and left, respectively, and the most statistically significant metabolites are towards the top. Heatmaps of metabolite expression profiles were displayed using Cluster3.0 and Java TreeView. Metabolite enrichment analysis was performed using MetaboAnalyst 5.0 [20] to determine whether a group of functionally related metabolites is significantly important. Data were visualized with double gradient heat map of GraphPad Prism version 7.0 program.

### 2.16. Blood Biochemistry

Each group of wild-type (*n* = 6) and Ndrg3 LKO (*n* = 6) male mice at the age of 8–12 weeks was used for blood analysis. Analysis of alanine aminotransferase (ALT), aspartate aminotransferase (AST), alkaline phosphatase (ALP), albumin and total protein, total bilirubin, and L-lactate dehydrogenase from blood was performed using a Hitachi 7020 automatic analyzer (Hitachi, Ltd., Tokyo, Japan).

### 2.17. Triglyceride (TG) Measurement

The triglyceride level was measured with a Colorimetric Assay Kit (Abcam, ab65336) following the manufacturer’s instruction. Triglyceride extracted from liver tissue was incubated with lipase to convert triglyceride to glycerol and free fatty acid for 20 min and reacted with a dye to generate a colored product by incubation with an enzyme mix for 30 min. Color absorbance was measured with a microplate reader (OD 570 nm).

### 2.18. RNA Sequencing

RNA was extracted from the same liver tissues used for metabolome analysis using an RNA Extraction kit (Qiagen, Venlo, Netherlands, #74104). RNA quality was checked with Agilent 2100 Bioanalyzer and total RNA (3 ug) satisfying RNA integrity number (RIN) > 8.0 was used. RNA sequencing was performed using an Illumina Stranded mRNA kit for library preparation with 100 bp Paired End reads. Sequenced reads were mapped to mouse genome (mm9) using STAR (v.2.5.1), and the gene expression levels were quantified with count module in STAR. Raw data were deposited with NCBI (GEO accession number; GSE196110). The Z-score for the expression of each gene was visualized with a double gradient heat map of GraphPad Prism version 7.0 program.

### 2.19. Statistical Analysis

Statistical significance was determined with two-tailed unpaired Student’s *t*-test for two groups, or two-way ANOVA with Tukey’s correction for multi-comparison. All graphs were depicted with GraphPad Prism version 7.0. Error bars display mean ± standard error of the mean (SEM). Quantification for immunoblot was performed using ImageJ software. Statistically significant *p*-values were taken below * *p* <0.05, ** *p* < 0.01, and *** *p* < 0.001 vs. control, respectively.

## 3. Results

### 3.1. Liver-Specific Ablation of NDRG3 Results in Excessive Hepatic Glycogen Accumulation and Liver Injury

To evaluate the possible metabolic roles of NDRG3 in liver, we established Ndrg3 LKO mouse lines via sequential mating of the mice carrying recombinant *Ndrg3* gene whose exon 4 region is flanked by loxP sites, i.e., first with mice expressing Flp recombinase, followed with mice expressing Cre recombinase in a liver-specific manner under the control of the albumin gene promoter (Appendix A). Liver-specific disruption of NDRG3 was validated at genomic DNA, transcriptional, and translational levels (Appendix A). Ndrg3 LKO mice showed two severe gross phenotypic defects, i.e., smaller body size than corresponding wild-type (WT) littermates likely reflecting growth retardation, and a high rate of premature lethality (30–40%) at the age of 8–12 weeks mostly in association with abdominal ascites (Appendix A). These phenotypes were observed in both males and females. In contrast, livers of Ndrg3 LKO mice were mostly bigger than those of WT in size and tended to display yellowish colors distinguished from the reddish brown color of WT livers (Appendix A). Accordingly, male Ndrg3 LKO mice at 10–12 weeks of age exhibited significantly higher liver/body weight ratios than corresponding WT mice (Figure 1A). However, food consumption was not significantly different between the two groups of mice (Appendix A).

H&E staining of histological sections of mouse liver showed extensive degenerative changes in Ndrg3 LKO mice involving swelling and ballooning of hepatocytes and widespread perinuclear vacuoles, which were progressively exacerbated with age (Figure 1B). Blood tests for biomarkers of liver function and injury in 2-month-old mice revealed signatures indicative of increased liver injury as well as lowered liver functions (Figure 1C). Examination of hepatic lipid accumulation by oil-red O staining indicated no significant difference between Ndrg3 LKO and WT (data not shown) but PAS staining revealed an extensive glycogen accumulation in Ndrg3 LKO livers as evidenced by magenta granules in cytoplasm, indicating that NDRG3 loss may cause abnormal glycogen accumulation in liver (Figure 1D). Quantitative analysis of glycogen levels in overnight-fasted male mice confirmed that unlike the relatively steady levels of hepatic glycogen seen in WT livers, glycogen volume in Ndrg3 LKO livers was progressively elevated in an age-dependent manner, showing a significant difference from that of WT after 2 months of age (Figure 1E). We then isolated hepatocytes from livers of 12-week-old male mice by collagenase perfusion, incubated overnight in Medium 199 containing 5 mM glucose, and assessed their glycogen content through PAS staining (Figure 1F). The primary hepatocytes from Ndrg3 LKO mice showed a much stronger PAS staining than WT hepatocytes, indicating that dysregulation of glycogen metabolism in Ndrg3 LKO livers is intrinsic to hepatocytes rather than due to a systemic effect following NDRG3 depletion in liver. In summary, disruption of NDRG3 in mouse liver causes aberrant glycogen accumulation, in association with severe hepatomegaly and liver injury, suggesting that NDRG3 might be involved in the regulation of important steps in glycogen metabolism in the liver.

### 3.2. Liver-Specific NDRG3 Ablation Causes Impairment of Hepatic Glucose Homeostasis

Impairment of hepatic glycogen metabolism can be one of main reasons to disrupt the systemic glucose homeostasis [21]. Having observed aberrant glycogen accumulation in Ndrg3 LKO livers and primary hepatocytes, we assessed blood glucose levels in Ndrg3 LKO mice. The blood glucose levels of Ndrg3 LKO males at 8–12 weeks of age were significantly lower than those of WT mice both ad libitum and at an overnight-fasted condition (Figure 2A). Similar observations were obtained with female mice (Appendix A). Ndrg3 LKO mice exhibited much faster clearance of exogenous glucose than WT in GTT, with the AUC of overall glucose tolerance decreased by about 40% (Figure 2B). These observations suggest that hepatic NDRG3 ablation might severely hinder the flux of metabolic pathways regulating the mobilization/storage of glucose in liver.

G6P and G1P are essential intermediates of the metabolic pathway for the interconversion between glycogen and free glucose. We found that free glucose levels were highly reduced in Ndrg3 LKO livers compared to those in WT (Figure 2C), but G6P and G1P levels were significantly upregulated in Ndrg3 LKO livers (Figure 2D,E). These results indicate that the glucose–glycogen flux is highly shifted toward glycogen accumulation in Ndrg3 LKO livers (Figure 2F), suggesting aberrant glycogen metabolism as the cause of hypoglycemia observed in Ndrg3 LKO mice.

### 3.3. Glycogen Synthesis Is Not Hyper-Activated in Ndrg3 LKO Mice

Shift of glucose–glycogen flux toward glycogen accumulation in Ndrg3 LKO livers could result from abnormalities in glycogen synthesis/breakdown. We first examined the glycogen synthesis capacity by assessing the expression of GS, a rate-limiting enzyme in the glycogen biosynthetic pathway. qRT-PCR analysis indicated that *Gys2* gene expression, the liver isozyme of GS, in overnight-fasted Ndrg3 LKO mouse livers was largely comparable to that of WT mice, with a decreasing tendency at a later age (Figure 3A), and this pattern was maintained in mice refed for 6 h (Appendix A). Western blot analysis indicated that GS protein levels were not significantly different between WT and Ndrg3 LKO livers at 1 month of age but then tended to decrease thereafter in Ndrg3 LKO livers compared to WT (Figure 3B and Appendix A). Levels of inhibitory phosphorylation of GS at Ser641 also showed a similar pattern (Appendix A). As a result, the ratio of p-GS(S641) to total GS was steadily maintained in both Ndrg3 LKO and WT livers regardless of age (Figure 3B). Insulin signaling plays a critical role in glycogen synthesis by positively regulating GS activity via AKT-glycogen synthase kinase 3 beta (GSK3β)-GS axis. Time-dependent profiles of insulin signaling in primary hepatocytes isolated from 2-month-old mice showed that insulin-induced AKT activity and its inhibitory phosphorylation of GSK3β at Ser9 were more or less comparable between Ndrg3 LKO and WT hepatocytes (Figure 3C). In addition, the p-GS(S641)/GS profile in insulin-stimulated hepatocytes was also similar between Ndrg3 LKO and WT mice, suggesting that insulin signaling may work normally or at least is not hyper-activated in Ndrg3 LKO hepatocytes. Finally, the enzymatic activity of GS measured from liver tissues of 2- and 4-month-old Ndrg3 LKO mice was not significantly different from that of WT (Figure 3D). Collectively, these data indicate that glycogen synthesis capacity is not excessively elevated in Ndrg3 LKO liver but rather comparable to that in WT liver.

### 3.4. NDRG3 Abrogation in Liver Suppresses Glycogen Degradation Due to DownRegulation of PYGL and Glucose-6-Phosphate Transporter Member 4 (G6PT; SLC37A4) Expression

Since glycogen synthesis activity was not particularly higher in Ndrg3 LKO livers and hepatocytes than in WT counterparts, we next investigated the regulation of glycogen breakdown. We examined the dynamics of glycogen turnover dependent on the feeding conditions in mice, i.e., fasting and refeeding. At 2 months of age, hepatic glycogen volumes stayed low in a fasting condition in both WT and Ndrg3 LKO mice but were highly increased upon refeeding (Figure 4A). At 4 months of age, the glycogen volume of fasted WT mice remained low, but that of Ndrg3 LKO mice was maintained much higher in between those of fasted and refed WT mice, presenting a noticeable difference between Ndrg3 LKO and WT livers. However, this difference was not observed in refed animals. Since hepatic glycogen volume of Ndrg3 LKO mice progressively increased with age (Figure 1E), these data seem to indicate that fasting-associated glycogen breakdown might be dysregulated in aged Ndrg3 LKO livers (Figure 4A). We, therefore, quantitatively analyzed the expression of glycogen breakdown pathway genes in 2–4-month-old mouse livers at different feeding conditions to identify the ones showing a high dependence on NDRG3 for their expression (Figure 4B,C). We found that the expression of *Pygl* and *G6pt* was significantly downregulated in Ndrg3 LKO livers compared to WT in both fasted and refed conditions (Figure 4C). In contrast, debranching enzymes of the glycogenolysis pathway (*Agl* and *Gaa*) showed an expression pattern independent of NDRG3.

Since PYGL is the rate-limiting enzyme of fasting-associated glycogen degradation [22], we further examined its expression at the protein level. PYGL protein expression in primary hepatocytes from 3-month-old mice was highly downregulated in Ndrg3 LKO relative to WT, indicating that PYGL downregulation in Ndrg3 LKO mice is intrinsic to hepatocytes (Figure 4D). We then assessed PYGL protein expression in 2- and 4-month-old mice depending on feeding conditions. The amount of total PYGL protein expressed in 2-month-old livers remained relatively constant regardless of the feeding condition or the *Ndrg3* genotype (Figure 4E and Appendix A). The level of catalytically active PYGL with Ser15 phosphorylation (p-PYGL(S15)) in 2-month-old livers decreased upon feeding in both mouse groups and was maintained relatively lower in Ndrg3 LKO livers than in WT. In 4-month-old mice, feeding-dependent changes in total PYGL and p-PYGL(S15) levels were not evident in WT livers but both proteins were significantly downregulated in Ndrg3 LKO livers compared to WT regardless of feeding conditions (Figure 4E and Appendix A). When the PYGL and p-PYGL(S15) profiles were compared with the glycogen profiles at different feeding conditions in Figure 4A, 4-month-old mice in the fasted condition appeared to show the best matches between the glycogen profile and the profiles of PYGL and p-PYGL(S15). These results suggest that NDRG3 is required for the proper operation of the fasting-associated glycogen degradation in aged mouse livers as its deficiency can significantly hamper PYGL expression and activity.

PYGL activity is highly regulated at the protein level through post-translational modifications involving serine phosphorylation and/or by binding of allosteric effectors such as G6P, ATP, and AMP [23]. Expression of protein phosphatase 1 (PP1), which dephosphorylates p-PYGL(S15), was not different between WT and Ndrg3 LKO livers (Appendix A). However, expression of phosphorylase kinase regulatory subunit alpha 2 (*Phka2*), which regulates phosphorylase kinase to activate PYGL by serine phosphorylation, was significantly downregulated in fasted Ndrg3 LKO relative to WT (Figure 4C). In addition, the G6P level was significantly upregulated in Ndrg3 LKO livers (Figure 2D). Consistent with these observations, PYGL activity determined from liver extracts of 4-month-old mice was significantly diminished in Ndrg3 LKO livers compared to WT (Figure 4F). Collectively, these results indicate that the malfunctioning of PYGL due to the downregulation of gene expression and suppression of enzymatic activity in hepatocytes could be the primary cause of aberrant hepatic glycogen accumulation and hypoglycemia in Ndrg3 LKO mice.

### 3.5. Liver-Specific Abrogation of NDRG3 Suppresses Hepatic Gluconeogenesis but Stimulates Liver Triglyceride Accumulation

Under fasting conditions, the liver initially increases glycogenolysis in response to glucagon, and then, during a longer term fasting, gluconeogenesis takes the role [21]. In our results describing age-dependent glucose–glycogen flux, it is notable that at 2 months of age, Ndrg3 LKO mice exhibited hypoglycemia (Figure 2A) and lower intrahepatic glucose levels (Figure 2C) without showing significant changes in fasting hepatic glycogen content compared to WT (Figure 4A). This observation suggests that at this stage of development hypoglycemia might be caused in Ndrg3 LKO mice by factors other than impairments in glycogenolysis. We, therefore, examined glucagon-induced gluconeogenic activities using primary hepatocytes from 2-month-old mice. Ndrg3 LKO hepatocytes gave a significantly lower amount of glucose production in response to glucagon than WT hepatocytes (Figure 5A). In contrast, the glucagon-induced glycogenolytic activity was comparable between the two groups of hepatocytes (Appendix A). qRT-PCR analysis of gluconeogenic gene expression in the livers of 2-month-old, fasted mice showed that *Pck1* (phosphoenolpyruvate carboxykinase) and *Fbp1* (fructose-1,6-bisphosphatase 1) were significantly downregulated in Ndrg3 LKO compared to WT (Figure 5B). In contrast, the levels of G1P and G6P were already upregulated at this stage in Ndrg3 LKO livers relative to those in WT livers (Figure 2D,E), likely due to *G6pt* downregulation as shown in Figure 4C. Therefore, at 2 months of age, the malfunctioning of gluconeogenesis appears to make a significant contribution to the development of hypoglycemia in Ndrg3 LKO mice.

In addition to excessive hepatic glycogen accumulation and hypoglycemia, GSDs often exhibit aberrant lipid accumulation in the liver. We examined the triglyceride (TG) content of Ndrg3 LKO livers in comparison to that of WT. At 2 months of age, Ndrg3 LKO mice exhibited significantly higher levels of hepatic TG accumulation than WT in a fasting condition (Figure 5C). qRT-PCR analysis on lipogenic genes revealed that in this condition the expression of *Srebp1c* and *Acc1* was upregulated in Ndrg3 LKO livers compared to WT (Figure 5D). However, genes involved in fatty acid (FA) desaturation or TG synthesis from FA were not differentially expressed between the two groups. In the refeeding condition, the hepatic TG levels were comparable between the two mouse groups (Figure 5C). In addition, the highly upregulated expression of *Srebp1c*, *Acc1*, and *Fasn*, observed in WT in response to food intake, was much diminished in Ndrg3 LKO mice (Appendix A). These results indicate that Ndrg3 LKO mice exhibit increased hepatic FA synthesis and TG accumulation relative to WT in the fasting condition as a part of manifesting GSD phenotypes.

### 3.6. NDRG3 Depletion Perturbs Methionine Metabolism in Mouse Hepatocytes

We carried out metabolome profiling on mouse liver extracts via CE-TOFMS to quantitatively assess alterations in metabolite level downstream of NDRG3 ablation. Principal component analysis (PCA) and one-way hierarchical clustering analyses showed a clear separation of metabolome profiles for Ndrg3 LKO and WT livers (Appendix A). Sixty-nine metabolites satisfying (log_2_(fold change)>1, *p*-value < 0.05) in Ndrg3 LKO relative to WT were selected (Figure 6A and Appendix A) for metabolite set enrichment analysis (MSEA) using the MetaboAnalyst 5.0 program [20] (Figure 6B). Among top-ranked metabolite sets, those related to ‘methionine’ metabolism were highly enriched, including metabolomes of ‘methionine metabolism’, ‘glycine and serine metabolism’, ‘spermidine and spermine biosynthesis’, and ‘homocysteine degradation’ that were significantly upregulated in Ndrg3 LKO livers as well as the metabolomes of ‘methionine metabolism’ and ‘betaine metabolism’ that were significantly downregulated (Figure 6B, Appendix A). Methionine plays important roles in many cellular processes including transmethylation reactions, redox maintenance, polyamine synthesis, and coupling to folate metabolism via its involvement in one-carbon metabolism by way of the methionine cycle [24] (Figure 6C). We, therefore, focused on the metabolomic changes in the methionine cycle and its branch pathways. Among others, most of the metabolites in the ‘methionine cycle’ exhibited significant quantitative changes in Ndrg3 LKO livers relative to WT (Figure 6D). Thus, methionine and S-adenosylmethionine (SAM) were significantly upregulated in Ndgr3 LKO livers, while sarcosine, S-adenosylhomocysteine (SAH), adenosine, and dimethylglycine (DMG) were significantly downregulated. In addition, several metabolites in the branch pathways of the methionine cycle, i.e., methylthioadenosine (MTA; methionine salvage pathway), cystathionine (transsulfuration pathway), and glutathione (GSH) and GSSG (glutathione synthesis pathway), were significantly upregulated (Figure 6C,D). These results indicate that a major perturbation has occurred in the methionine cycle in Ndrg3 LKO livers, causing a redirection of its flux towards the branch pathways.

### 3.7. NDRG3 Depletion Causes Restructuring of the Methionine Cycle in Mouse Hepatocytes

Next, we analyzed the mouse livers used in metabolome analysis by RNA sequencing to examine whether the metabolomic alterations in the methionine cycle and its branch pathways in Ndrg3 LKO livers were consistent with the relevant gene expression profiles. Consistent with the disturbances in metabolite levels in the methionine cycle, expression of constituent genes (glycine N-methyltransferase (*Gnmt*), adenosylhomocysteinase (*Ahcy*), and betaine-homocysteine S-methyltransferase (*Bhmt*)) was highly suppressed in Ndrg3 LKO livers except for methionine adenosyltransferase 1 (*Mat1*) gene encoding the enzyme for the 1st step of the cycle (Figure 7A). In contrast, expression of branch pathway genes, cystathionine beta-synthase (*Cbs*) and glutathione synthetase (*Gss*), which are responsible for the production of cystathionine and GSH, respectively, was significantly upregulated in Ndrg3 LKO livers. We combined the gene expression data with metabolome data in the pathway map of the methionine cycle and branch pathways and examined the compatibility between the metabolomic *versus* genomic changes caused by NDRG3 depletion in the liver (Figure 7B). Downregulation of *Gnmt* appeared to be consistent with the accumulation of its substrate, SAM, which in turn seemed to result in the upregulation of MTA in the branch pathway. The expression pattern of transsulfuration pathway genes was consistent with the accumulation of cystathionine as *Cbs* was upregulated whereas cystathionase (*Cth*) was downregulated (Figure 7A,B). Cystathionine production might be further promoted by SAM upregulation as it can allosterically activate CBS enzyme activity [25], and by the downregulation of *Bhmt*, which facilitates the channeling of its substrate (homocysteine) to the transsulfuration pathway. The upregulated cystathionine production seems to result in increased GSH production in Ndrg3 LKO livers owing to the upregulation of *Gss* (Figure 7A,B). Therefore, the genomic changes caused by NDRG3 depletion in mouse hepatocytes seem to reasonably coincide with the metabolomic changes. Collectively, these results indicate that the cyclic nature of the methionine cycle was disrupted at the genomic level in Ndrg3 LKO livers, and the methionine metabolism was restructured to cause a significant accumulation of several metabolites in the branch pathways of the methionine cycle (MTA, cystathionine, and GSH).

### 3.8. GNMT DownRegulation Is a Crucial Upstream Event in the Reprogramming of the Methionine Cycle in Ndrg3 LKO Livers

We finally analyzed the NDRG3 dependence of gene expression in the methionine cycle and branch pathways to identify the ones most directly implicated in the restructuring of the methionine cycle in NDRG3-deficient liver. qRT-PCR analysis on mouse livers indicated that expression of the methionine cycle genes, *Gnmt* and *Bhmt*, was highly dependent on *Ndrg3* expression as their mRNA levels were significantly downregulated in Ndrg3 LKO livers compared to WT in both fasted and refed conditions (Figure 8A). Expression patterns of the other genes of the methionine cycle (*Ahcy*, *Mat1*) or branch pathways (*Cbs*, *Mtap*) were not dependent on *Ndrg3*, particularly in the refed condition. Since the mRNA expression and enzymatic activity of BHMT is known to be feedback-inhibited by SAM [26], and since GNMT ablation is known to cause SAM upregulation in mice [27], GNMT downregulation is likely at the upstream of BHMT downregulation in Ndrg3 LKO livers. Western blot analysis showed that GNMT protein expression is also significantly downregulated in Ndrg3 LKO livers relative to WT in an age-dependent manner (Figure 8B and Appendix A). A similar pattern of GNMT downregulation was observed in NDRG3-deficient primary hepatocytes, indicating that suppression of GNMT expression in Ndrg3 LKO livers is intrinsic to hepatocytes (Figure 8C). In summary, these results suggest that GNMT downregulation and accompanying SAM elevation might play crucial roles in the restructuring of the methionine cycle in Ndrg3 LKO livers.

## 4. Discussion

We investigated the pathophysiological roles of NDRG3 in relation to cell metabolism by generating a mouse model with a liver-specific ablation of the *Ndrg3* gene. These mice developed a number of pathologic phenotypes frequently observed in GSDs, including aberrant hepatic glycogen accumulation, hypoglycemia, TG accumulation, and hepatomegaly. Metabolome and transcriptome profiling indicated that glycogen metabolism as well as the methionine cycle was highly dysregulated in Ndrg3 LKO livers, suggesting that NDRG3 might play important roles in liver function and homeostasis by regulating cell metabolism at the upstream of glucose–glycogen metabolism and methionine metabolism.

We showed that hepatic glycogen volume was progressively increased in Ndrg3 LKO mice, in association with a decrease in glycogen degradation capability. Among the genes involved in the glycogen degradation pathway leading to free glucose generation, expression of *PYGL* and *G6PT* was significantly downregulated in Ndrg3 LKO livers in an *NDRG3*-dependent manner (Figure 4C). PYGL is a rate-limiting enzyme of glycogen breakdown converting glycogen into G1P [20], whereas G6PT facilitates the translocation of G6P from cytoplasm into ER lumen for subsequent hydrolysis to glucose and phosphate and thereby controls glucose/G6P concentration in the cytoplasm [28]. These genes are functionally linked with each other since G6PT downregulation causes elevation of G6P level in cytoplasm, inactivating PYGL [29]. The G6P level was significantly elevated in Ndrg3 LKO livers, and, combined with the downregulation of *Pygl* mRNA expression, overall PYGL enzymatic activity was significantly repressed in NDRG3-deficient hepatocytes compared to WT cells (Figure 4F). Deficiency in *PYGL* and *G6PT* genes is known to manifest as hypoglycemia and excessive glycogen accumulation, causing GSD VI [30] and GSD Ib [28], respectively, in humans. KO of *Pygl* or *G6pt* in mice was also shown to induce GSD phenotypes, including aberrant hepatic glycogen accumulation, hypoglycemia, and hepatomegaly [22,31]. These findings suggest that the GSD phenotypes observed in Ndrg3 LKO mice might have been caused by the downregulation of PYGL and G6PT expression/activity. Expression of PYGL and G6PT was also shown to be significantly downregulated in *Gnmt* KO mice [27]. GNMT expression is significantly downregulated in Ndrg3 LKO livers, and both *Gnmt* KO and Ndrg3 LKO mice share the forementioned GSD phenotypes with *Pygl* KO and *G6pt* KO mice. Therefore, it seems to be a good possibility that GNMT downregulation might be an important upstream event associated with the GSD phenotype manifestation in Ndrg3 LKO mice.

As indicated by the results of MSEA and gene expression analysis, alterations in the flux of the methionine cycle were outstanding in Ndrg3 LKO livers. Three out of the four steps in the methionine cycle were significantly downregulated except for the 1st step reaction, causing significant accumulation of SAM and methionine (Figure 6D and Figure 7B). The upregulated SAM seems to induce a number of notable changes in the flux of the methionine cycle in NDRG3-deficient hepatocytes. First, SAM accumulation appears to promote the accumulation of MTA in the branch pathway of the methionine cycle (Figure 6D and Figure 7B). Second, SAM is known to strongly inhibit the expression and activity of BHMT, and thus suppresses remethylation of homocysteine back to methionine [32]. Third, SAM is known to allosterically activate CBS that catalyzes the 1st step of the transsulfuration pathway converting homocysteine to cystathionine [25]. Moreover, expression of *Cbs* is transcriptionally upregulated in Ndrg3 LKO livers. The elevated catalytic capacity of CBS, along with the suppression of the immediate downstream reaction due to downregulation of *Cth* expression, promotes significant accumulation of cystathionine in NDRG3-deficient hepatocytes (Figure 6D and Figure 7B). Therefore, aberrant SAM elevation appears to function as an important contributor to the restructuring of the methionine cycle in Ndrg3 LKO livers (Figure 7B and Figure 8D). High levels of SAM accumulation were reported in several studies on *Gnmt* KO in mice [27,33,34], demonstrating a causal role for GNMT deficiency in SAM elevation. Therefore, Ndrg3 LKO-dependent downregulation of GNMT expression appears to be intimately linked to both the reprogramming of the methionine cycle and the manifestation of GSD phenotypes in Ndrg3 LKO mice. However, it is not known how NDRG3 depletion suppresses GNMT expression in mouse hepatocytes and how GNMT deficit is linked to the downregulation of PYGL and G6PT expression in Ndrg3 LKO mice in vivo. In mammals, GNMT is known to play major roles in the regulation of SAM levels, as documented by studies of humans with GNMT deficiency and mouse models with GNMT KO (Reviewed in [35]). Since SAM is required for many important biological processes, regulation of GNMT activity is considered critical for homeostasis [36]. Therefore, studies addressing the regulatory relationship between NDRG3 and GNMT and its biological significance may reveal interesting physiological and pathophysiological aspects of cell metabolism.

It appears that NDRG3 depletion in mouse liver can be a damaging condition. Thus, elevation of liver injury markers was observed in Ndrg3 LKO mice (Figure 1C). GNMT deficiency was also suggested as the cause of liver injury in the mouse model of *Gnmt* KO [27]. In addition, histopathologic abnormalities such as degenerative changes and perinuclear vacuoles in hepatocytes were shared in *Gnmt* KO [27] and Ndrg3 LKO mice. Interestingly, reprogramming of the methionine cycle in Ndrg3 LKO livers brings about the accumulation of several metabolites that are known to have defensive capacities against cellular damage such as SAM, MTA, cystathionine, and GSH (Figure 7). These metabolites are also significantly upregulated in *Gnmt* KO mice [33]. SAM is known, in addition to its well-known function as the methyl donor, to ameliorate liver injury in many animal models via its ability, for example, to raise GSH level, suppress tumor necrosis factor (TNFA) expression, and protect hepatocytes against apoptosis [34]. MTA is a thioether compound that can exert a direct antioxidant effect [37]. Although MTA does not accumulate in significant amounts in most cells [38], it was shown to have a protective effect on oxidative liver damage in animal models of liver injury [39,40]. Cystathionine was reported to have anti-inflammatory [41] and antiapoptotic functions [42,43], possess a scavenging function against superoxide radicals [44], and play a hepatoprotective role against endoplasmic reticulum stress-induced injury [43,45]. GSH is well-known for its roles in the homeostatic maintenance of cellular redox balance and as an antioxidant protecting cells from oxidative damage and the xenobiotic toxicities [46]. Thus, these results suggest that in the absence of NDRG3, the methionine cycle can be redirected in hepatocytes towards the augmentation of hepatoprotective capacities.

NDRG3 is a pro-tumorigenic member of the NDRG family in contrast to the other three members that are mainly associated with tumor suppressive functions [4]. Due to mutations of catalytic amino acid residues in α/β-hydrolase domain, NDRG3 is considered enzymatically nonfunctional and thought to function as a scaffold protein mediating growth-promotive signaling [3,47]. NDRG3 is frequently overexpressed in many cancers, associated with aggressive cancer phenotypes and a poor prognosis [4], and shown to exhibit a number of pro-tumorigenic characteristics including promotion of tumor growth, angiogenesis, and metastasis (See Introduction). Dysregulation of growth signaling due to genetic or environmental causes is often accompanied by reprogramming of cell metabolism [48]. Both the methionine cycle and glucose–glycogen flux play essential roles in cell growth by supporting nucleotide biosynthesis via coupling to folate metabolism and energy metabolism, respectively. It seems possible, then, that dysregulation of growth signaling due to NDRG3 depletion could cause the reprogramming of the methionine cycle and glucose–glycogen flux in Ndrg3 LKO liver.

## 5. Conclusions

NDRG3 depletion in liver results in the manifestation of GSD phenotypes in mice due to impaired hepatic glucose homeostasis associated with the suppression of glycogenolysis and gluconeogenesis. NDRG3 deficiency in liver also causes restructuring of the methionine cycle due to the downregulation of GNMT expression and elevation of SAM levels. The Ndrg3 LKO-dependent GNMT downregulation seems to play a crucial role in the pathogenesis of Ndrg3 LKO livers. Our results suggest that NDRG3 may have important regulatory roles in liver cell metabolism at the upstream of glucose–glycogen flux via the regulation of methionine metabolism, and that its regulation may have a therapeutic potential for disorders with a dysregulation in glucose–glycogen metabolism and methionine metabolism.

## Figures and Tables

**Figure 1 cells-11-01536-f001:**
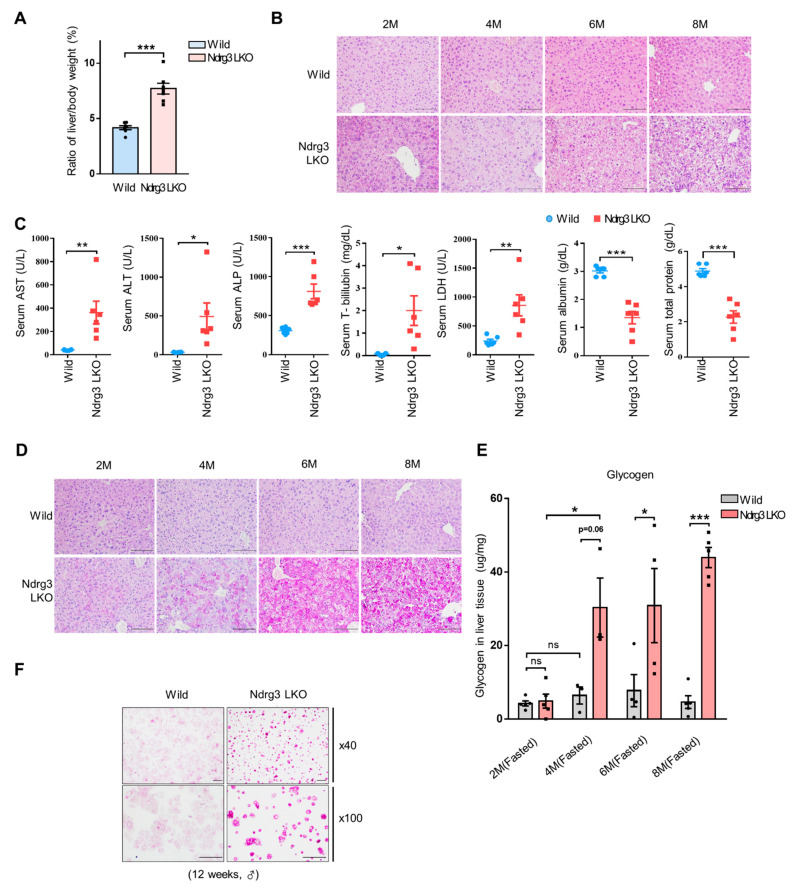
Pathologic changes and abnormal glycogen accumulation in NDRG3-depleted mouse livers. (**A**) Liver-to-body weight ratios of wild-type and Ndrg3 LKO mice at the age of 8–12 weeks (*n* = 7 per group). (**B**) Hematoxylin and eosin stained liver tissues of wild-type and Ndrg3 LKO mice at increasing ages. Scale bars; 100 μm. (**C**) Quantification of liver functional parameters analyzed from blood plasma of 2-month-old wild-type and Ndrg3 LKO mice (*n* = 6 per group). ALT, alanine transaminase; AST, aspartate transaminase; ALP, alkaline phosphatase; T-bilirubin, total bilirubin; LDH, lactate dehydrogenase. (**D**) Periodic acid-Schiff (PAS) staining of liver tissues from overnight-fasted wild-type and Ndrg3 LKO mice. Scale bars; 100 μm. (**E**) Quantification of hepatic glycogen volume in wild-type and Ndrg3 LKO mice at increasing ages. (*n* = 3–5 per group). (**F**) Representative PAS staining images of primary hepatocytes isolated from the livers of 12-week-old male wild-type and Ndrg3 LKO mice. Scale bars; 200 μm. Data are presented as mean ± SEM. *p*-values were calculated using two-tailed Student’s *t*-test (**A**,**C**), or two-way ANOVA with Tukey’s correction for multi-comparison (**E**). * *p* < 0.05, ** *p* < 0.01, and *** *p* < 0.001 vs. wild mice.

**Figure 2 cells-11-01536-f002:**
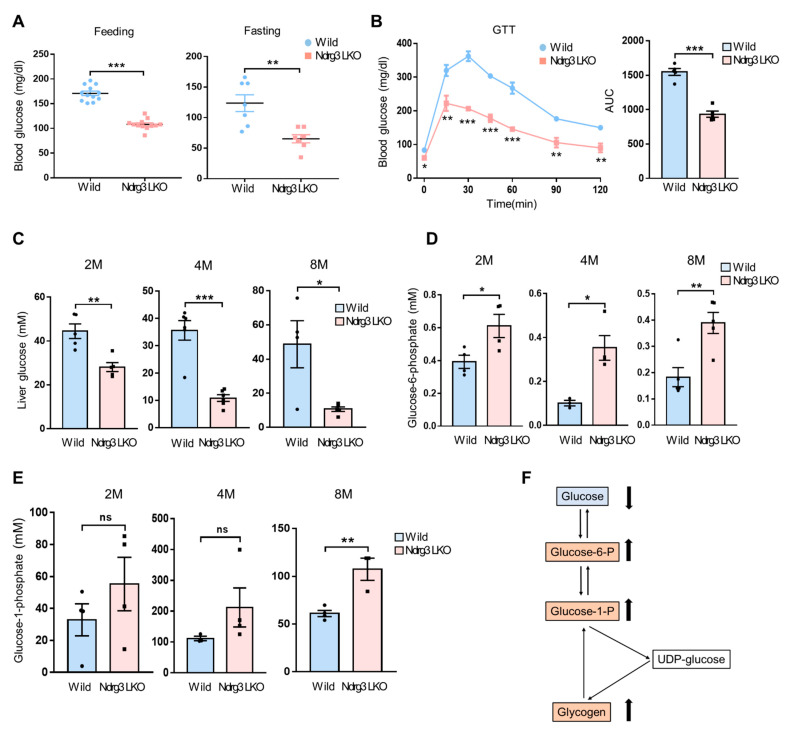
Hypoglycemia and impaired hepatic glucose homeostasis caused by liver-specific NDRG3 ablation. (**A**) Blood glucose levels of 2–4-month-old wild-type and Ndrg3 LKO mice determined ad libitum (Feeding, *n* = 12) or after fasting for 15 h (Fasting, *n* = 7). (**B**) Glucose tolerance test (GTT) (left), and AUC (area under the curve) for GTT (right). Male mice were fasted overnight and then injected with glucose at a dose of 2 g/kg body weight by intraperitoneal administration (*n* = 5 per group). (**C**–**E**) Levels of the metabolic intermediates in glycogen degradation pathway measured from liver tissues of wild-type and Ndrg3 LKO mice at increasing ages. Free glucose (**C**), glucose-6-phosphate (G6P; **D**), and glucose-1-phosphate (G1P; **E**). (**F**) Scheme for the hepatic glycogen metabolism. Changes in metabolite levels in the liver of Ndrg3 LKO mice relative to wild-type are shown as arrows. Data are presented as mean ± SEM. *p*-values were calculated using two-tailed Student’s *t*-test (**A**,**C**–**E**). * *p* < 0.05, ** *p* < 0.01, and *** *p* < 0.001 vs. wild mice.

**Figure 3 cells-11-01536-f003:**
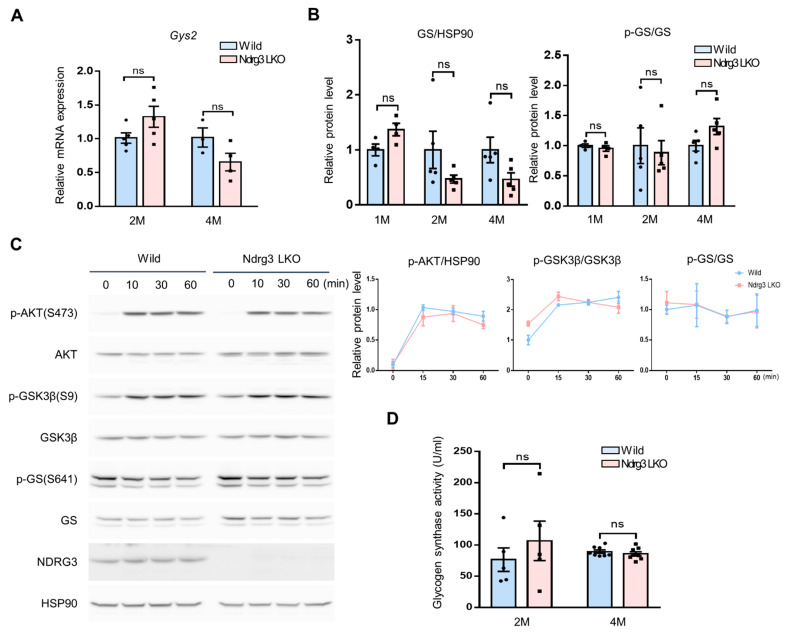
Intact glycogen biosynthesis capacity in Ndrg3 LKO mice. (**A**) qRT-PCR analysis of *Gys2* gene expression in overnight-fasted wild-type and Ndrg3 LKO mouse livers at 2 or 4 months of age (*n* = 3–4 per group). Gene expression was normalized using *Gapdh* as the reference. (**B**) Expression of glycogen synthase (GS) protein in wild-type and Ndrg3 LKO mouse livers at increasing ages (*n* = 4–5 per group). Total GS and p-GS(S641) were quantified from immunoblots shown in Appendix A using ImageJ program. The levels of p-GS(S641) relative to total GS (p-GS/GS) are shown on the right. (**C**) Time-dependent analysis of insulin signaling in primary hepatocytes obtained from 2-month-old wild-type and Ndrg3 LKO mice. Insulin signaling was assessed by Western blotting using p-AKT(S473), p-GSK3β(S9), and p-GS(S641) as the markers. (**D**) GS activity in the livers of 2-month-old wild-type and Ndrg3 LKO mice (*n* = 5 per group).

**Figure 4 cells-11-01536-f004:**
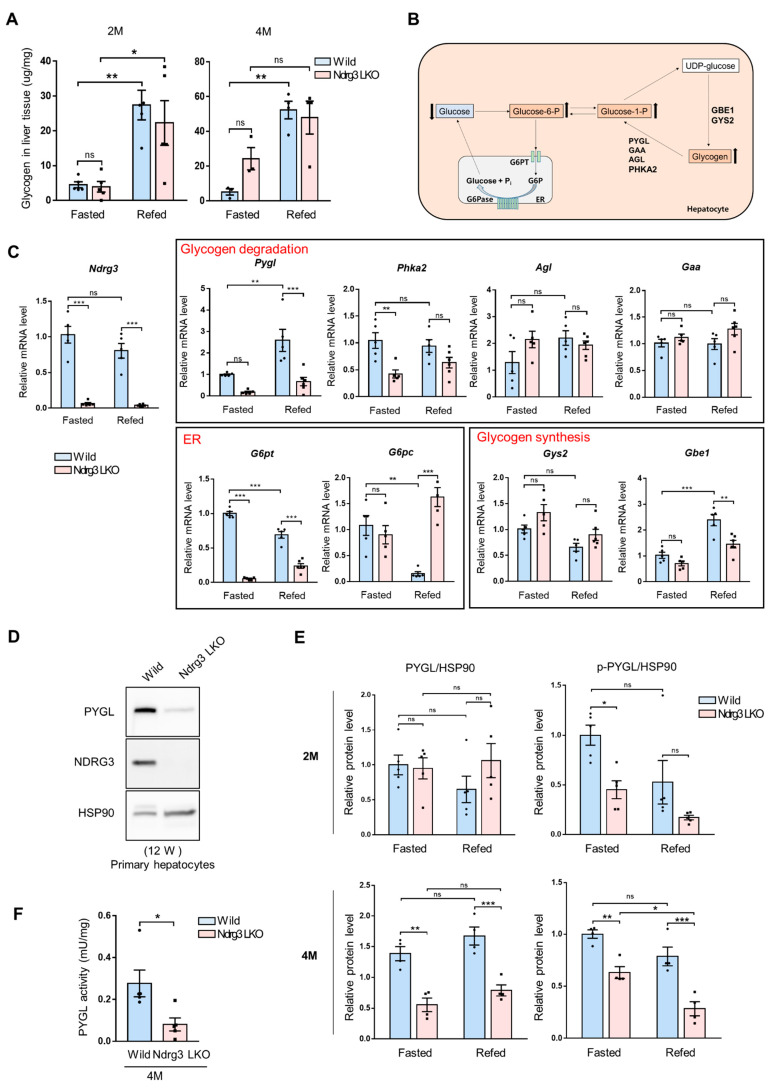
Impaired glycogenolysis in Ndrg3 LKO livers due to dysregulation of genes in glycogen degradation pathway. (**A**) Quantitative measurement of hepatic glycogen volumes depending on feeding conditions in 2- and 4-month-old wild-type and Ndrg3 LKO mice (*n* = 3–5 per group). Mice were fasted for 15 h or refed for additional 6 h before being subject to the glycogen analysis. (**B**) Schematic depiction of representative genes and pathways involved in the mobilization and/or storage of glucose in liver. Genes whose deficiency is implicated in the manifestation of glycogen storage disease phenotypes in humans are shown. (**C**) qRT-PCR analysis of the expression of glycogen metabolism pathway genes in the livers of wild-type and Ndrg3 LKO mice at 2 months of age. Mice were fasted for 15 h or refed for additional 6 h before being subject to the gene expression analysis (*n* = 4–5 per group). (**D**) PYGL protein expression in primary hepatocytes isolated from 3-month-old wild-type and Ndrg3 LKO mice. (**E**) Expression of PYGL and p-PYGL(S15) in wild-type and Ndrg3 LKO mouse livers at the age of 2 months (*n* = 5 per group) or 4 months (*n* = 4 per group). Mice were fasted for 15 h or refed for additional 6 h before being subject to the Western blot analysis. Total PYGL and p-PYGL(S15) were quantified from immunoblots shown in Appendix A using ImageJ program. (**F**) PYGL activity measured from the livers of 4-month-old wild-type and Ndrg3 LKO mice (*n* = 5 per group). Mice were fasted for 15 h and then refed for 6 h before being subject to PYGL activity assay. Gene expression was normalized using *Gapdh* as the reference. Data are presented as mean ± SEM. *p*-values were calculated using two-tailed Student’s *t*-test (**F**), or two-way ANOVA with Tukey’s correction for multi-comparison (**A**,**C**,**E**). * *p* < 0.05, ** *p* < 0.01, and *** *p* < 0.001 vs. wild mice.

**Figure 5 cells-11-01536-f005:**
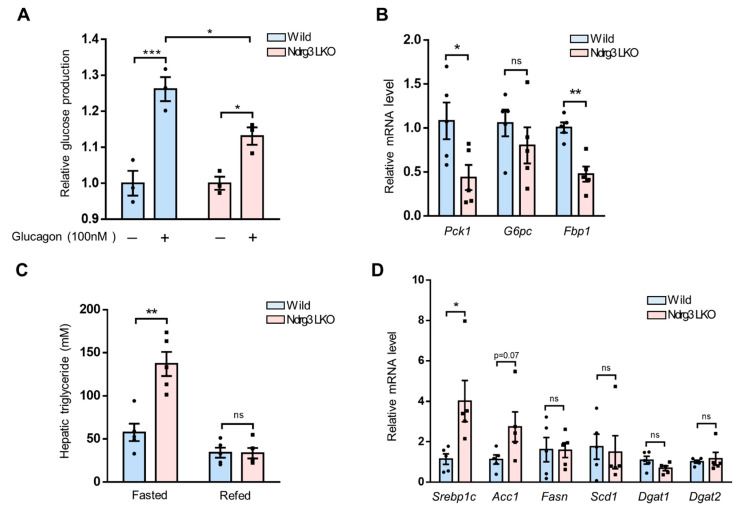
Alterations in hepatic gluconeogenesis and lipogenesis in Ndrg3 LKO mice. (**A**) Quantitative measurement of hepatic glucose production under a gluconeogenic condition. Primary hepatocytes isolated from 2-month-old wild-type and Ndrg3 LKO mice (*n* = 3 per group) were cultured, subjected to serum and glucose starvation overnight, and stimulated with glucagon (100 nM) for 4 h. Glucose production rate of glucagon-stimulated primary hepatocytes was expressed as the fold change relative to vehicle-treated hepatocytes. (**B**) qRT-PCR analysis of gluconeogenic gene expression in wild-type and Ndrg3 LKO mouse livers at 2 months of age in the fasting condition (*n* = 5 per group). (**C**) Liver triglyceride levels of 2-month-old wild-type and Ndrg3 LKO mice were determined (*n* = 5 per group). Mice were fasted overnight before being subject to the triglyceride analysis. (**D**) qRT-PCR analysis of lipogenic gene expression in wild-type and Ndrg3 LKO mouse livers at 2 months of age (*n* = 5 per group). Gene expression was normalized using *Rpl32* (**B**) or *Gapdh* (**D**) as the reference. Data are presented as mean ± SEM. p-values were calculated using two-way ANOVA followed by Tukey’s multiple comparisons test (**A**) or two-tailed Student’s *t*-test (**B**–**D**). * *p* < 0.05, ** *p* < 0.01, and *** *p* < 0.001 vs. wild-type mice.

**Figure 6 cells-11-01536-f006:**
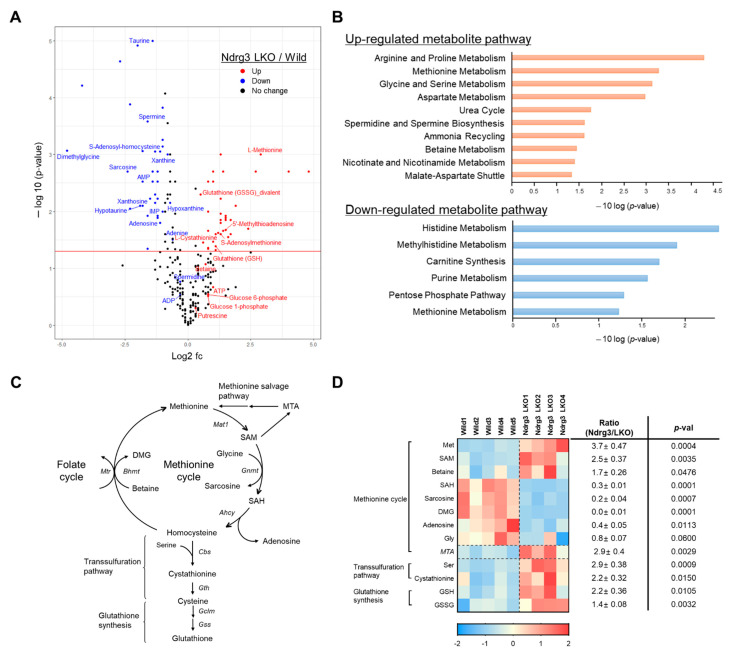
Perturbation of methionine metabolism caused by NDRG3 ablation in mouse hepatocytes. (**A**) A volcano plot for identifying differentially expressed metabolites between wild-type (*n* = 5) and Ndrg3 LKO mouse livers (*n* = 4). Blue dots with (*p* < 0.05, log_2_(fold change) < −1) indicate metabolites significantly downregulated in Ndrg3 LKO livers compared to wild-type, whereas red dots with (*p* < 0.05, log_2_(fold change) > 1) represent metabolites significantly upregulated in Ndrg3 LKO livers. Mice at 10–12 weeks of age were sacrificed under ad libitum states and used in the metabolome analysis. (**B**) Metabolome sets enriched in the livers of Ndrg3 LKO mouse relative to wild-type analyzed by MetaboAnalyst 5.0. Total of 68 metabolites satisfying (*p* < 0.05, log_2_(fold change) > 1) between Ndrg3 LKO and wild-type were subject to the enrichment analysis, and the metabolome sets with *p* < 0.05 are shown. Red and blue, up- and down-regulated in Ndrg3 LKO livers relative to wild-type, respectively. (**C**) A schematic diagram representing the methionine cycle and its branch pathways. (**D**) Heatmap and relative ratios for the expression of metabolites of the methionine cycle and its branch pathways in the livers of wild-type (*n* = 5) and Ndrg3 LKO (*n* = 4) mice. Red, relatively upregulated. Blue, relatively downregulated. Data are presented as mean ± SEM. *p*-values were calculated using two-tailed Student’s *t*-test (**D**).

**Figure 7 cells-11-01536-f007:**
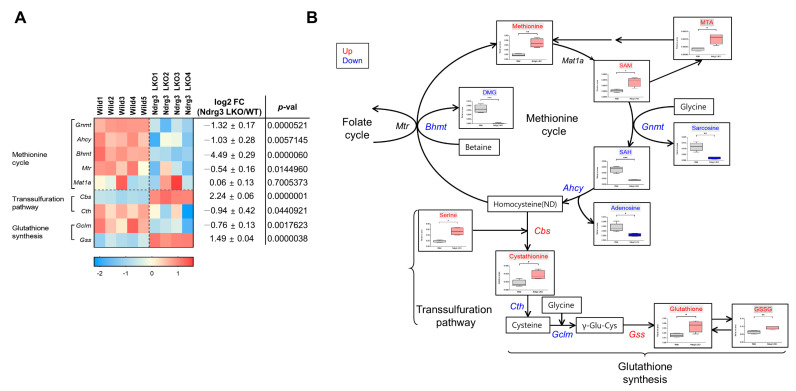
Restructuring of the methionine cycle in NDRG3-depleted mouse hepatocytes. (**A**) Heatmap and the relative expression of the genes of the methionine cycle and its branch pathways in the livers of wild-type (*n* = 5) and Ndrg3 LKO (*n* = 4) mice. Red, relatively upregulated. Blue, relatively downregulated. (**B**) An integrated diagram for the metabolic and genomic activities of the methionine cycle and its branch pathways. Genes and metabolites showing a significantly different expression between wild-type and Ndrg3 LKO livers are colored. Red and blue, significantly up- and down-regulated, respectively, in Ndrg3 LKO livers relative to wild-type. Gene expression data were obtained from RNA-Seq analysis of the mouse livers used in the metabolome analysis. Relative expression of these metabolites and genes can also be found from heatmaps in Figure 5D and Figure 6A, respectively. Data are presented as mean ± SEM. *p*-values were calculated using two-tailed Student’s *t*-test. * *p* < 0.05, ** *p* < 0.01, and *** *p* < 0.001 vs. wild mice.

**Figure 8 cells-11-01536-f008:**
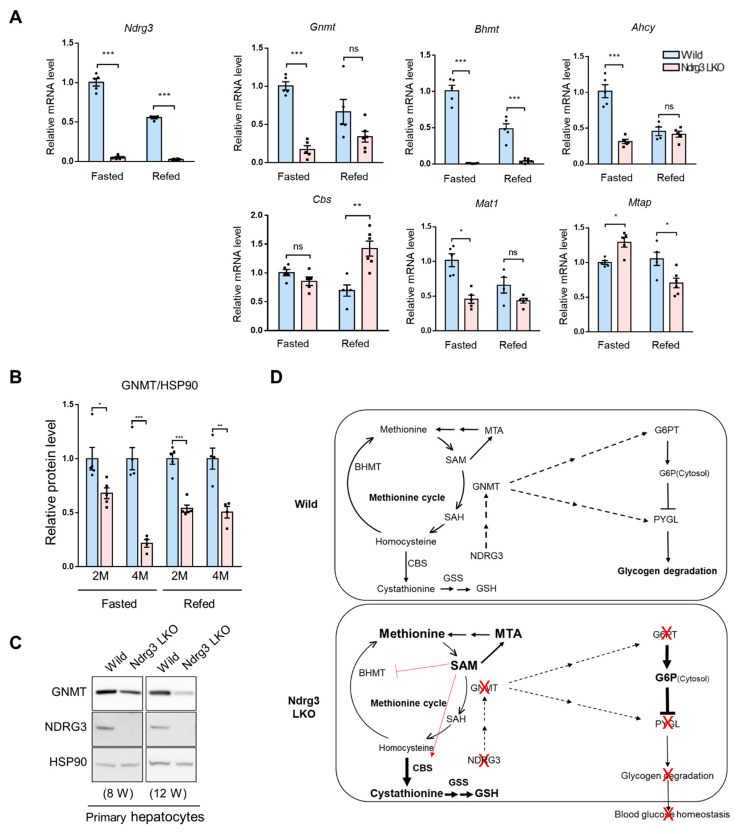
Critical role of GNMT downregulation in the reprogramming of the methionine cycle in Ndrg3 LKO livers. (**A**) qRT-PCR analysis of the expression of genes of the methionine cycle and branch pathways in the livers of wild-type and Ndrg3 LKO mice at 8-10 weeks of age. Mice were fasted for 15 h or refed for additional 6 h before being subject to the gene expression analysis (*n* = 4–5 per group). (**B**) GNMT protein expression in the livers of wild-type and Ndrg3 LKO mice. Mice at the age of 2 months (*n* = 5 per group) and 4 months (*n* = 4 per group) were fasted for 15 h or refed for additional 6 h before being subject to the protein expression analysis. (**C**) Western blot analysis of GNMT expression in primary hepatocytes isolated from wild-type and Ndrg3 LKO mice at 2 or 3 months of age. (**D**) Illustrative scheme describing plausible molecular mechanisms for the induction of methionine cycle reprogramming and GSD phenotypes in Ndrg3 LKO livers. Activation of CBS and inhibition of BHMT by SAM are indicated by red lines. Gene expression was normalized using *Gapdh* as the reference. Data are presented as mean ± SEM. *p*-values were calculated using two-tailed Student’s *t*-test. * *p* < 0.05, ** *p* < 0.01, and *** *p* < 0.001 vs. wild mice.

## Data Availability

The data presented in this study are included in the article/Appendix A. Raw data for RNA sequencing of mouse livers were deposited with NCBI (GEO accession number; GSE196110). Further inquiries can be directed to the corresponding author.

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
