# Peer review of "Glycogen Storage Disease Phenotypes Accompanying the Perturbation of the Methionine Cycle in NDRG3-Deficient Mouse Livers"

_cells, 2022, doi:10.3390/cells11091536_

Round 1
Reviewer 1 Report
This manuscript reports the metabolic consequences of the loss of NDRG3 specifically in the liver in a mouse model. The authors showed defects in hepatic glucose metabolism and perturbation of the methionine cycle. The authors showed a decrease gluconeogenesis and glycogenolysis that they attributed to decreased expression of glycogen phosphorylase and glucose-6-phosphate transporter. Using metabolomic and RNA sequencing, the authors showed a decrease in the expression of glycine N-methyltransferase (GNMT) which was accompanied with restructuration of methionine metabolism. They finally proposed that the loss of NDRG3 may promote hepatoprotective capacities.
There are various issues relating to the methodology and to the conclusion raised from the results that are of major concerns. The authors observed a decrease in GNMT expression in the liver of NDRG3 KO mouse but did not propose a regulatory mechanism. The link between GNMT and gluconeogenesis has already been well described and published and is not cited (Hughey et al. 2018, doi.org/10.1074/jbc.RA118.002568). In addition, the comparison of the liver NDRG3 KO mouse to GSD model is overstated.
- The comparison of the liver NDRG3 KO mouse to a glycogen storage disease model is overstated. GSD diseases are characterized by hypoglycaemia and high glycogen levels in the fed and fasted states. For example, GSD1b mouse (G6pt-/- mouse) exhibited 30mg of glycogen per g of liver (Raggi et al. 2018, doi: 10.1007/s10545-018-0211-2). The glycogen levels measured in the manuscript are very low : did the authors use a biochemical method to validate the Glycogen Assay Kit ? Can the authors precise the fasting state of the mouse studied in Fig 1E ?
- The authors cannot conclude on the effect of NDRG3 loss on lipid accumulation from oil-red staining but have to measure liver TG content.
- The quality of the images from Figure 1D are too low.
- Liver NDRG3 KO mice exhibited a strong decrease in fed and fasting blood glucose levels. However, mice lacking hepatic gluconeogenesis are still able to maintain normal blood glucose levels in fed and fasting states (Mutel et al. 2011, doi: 10.2337/db11-0571). Can the authors provide an explanation ?
- The visual comparison leading to the conclusion of the “more or less comparable” P-AKT levels is not appropriate (Fig 3C). Quantification has to be made to draw conclusions.
- The amount of P-PYGL has to be compared to PYGL and not HSP90 in Fig 4E to conclude on the phosphorylation state of the enzyme.
- Can the authors explain why the P-PYGL levels are completely different in the same conditions (4M fasted NDRG3 mice) in Figure S4A and S4B ?
- The authors concluded on the “hepatoprotective capacities” developed in the liver of NDRG3 KO mouse (line 647). Increased ASAT and ALAT activities and increased glycogen levels are shared between Ndrg3 LKO and GnmtKO (as the authors claimed in line 602). These parameters are usually not associated with protection of the liver physiology. Indeed, Gnmt KO mice are known to develop hepatocellular carcinoma (Martínez-Chantar et al. 2008).
Minor : The supplemental file did not contain Figure S5. The original blot of Figure 4E are lacking
Reviewer 2 Report
- The authors investigated the pathophysiological roles of NDRG3 in relation to cell metabolism by generating a mouse model with a liver-specific knockout of NDRG3 gene. In the time-dependent analysis of insulin signaling (Figure. 3C), western blotting was performed using 2-month-old mice, and they concluded that there was no significant difference in GS activity between Ndrg3 LKO and WT mice. Glycogen synthesis in Ndrg3 LKO liver did not increase significantly, which was similar to that in WT mice. I notice that the glycogen levels in Ndrg3 LKO livers was progressively elevated in an age-dependent manner, showing a significant difference from that of WT after 2 months of age (Figure 1E). Maybe the glycogen synthesis ability of the 2-month-old mice has not been activated, and glycogen significantly increased after 4 months. It would be more convincing to perform GS enzyme test in mice aged 4 months or older to exclude the influence of Ndrg3 depletion in glycogen synthesis.
- In Figure 4A, the difference in glycogen accumulation between the Ndrg3 LKO and WT mice was only observed under the fasting condition in 4-month-old mice. The authors speculated that the glycogen degradation in the older mice was disrupted under the fasting state, that is, the glycogen level in the Ndrg3 LKO group was increased. I wonder why the glycogen accumulation in the Ndrg3 LKO group was lower than WT after refeeding in 4-month-old mice? Does feeding activate glycogen degradation?
- For metabolome analysis, liver samples extracted from WT and Ndrg3 LKO mice at the age of 8-10 weeks. Why not 4-month-old mice?
- Many words in Figure 5 A and B are not clear enough.
Reviewer 3 Report
cells-1641670
This work by Dr. Sohn and colleagues aimed to understand the pathophysiological roles of NDRG3 in relation to cell metabolism, specifically in hepatocytes. The work appears technically sound and the manuscript is well-written. Several key issues have not been addressed and/or should be clarified to further strengthen the authors’ conclusions and proposed mechanisms.
Comments to the authors
-Hepatocyte-specific NDRG3 knockout mice showed hypoglycaemia in tandem with hepatic glucose-1/6-phosphate/glycogen accumulation, lower mRNA expression levels of Slc37a4 (Fig. 4C) and reduced expression and activity of glycogen phosphorylase (Fig. 4C-F). These data suggest reduced hepatic glucose production. Actual measures of hepatic glucose production (e.g. stable isotope-based rate of glucose appearance, pyruvate tolerance tests or glucose production rates by primary hepatocytes) are however lacking and should be included to connect the biochemical phenotype with the changes in enzyme/transporter expression.
-At 2 months of age, hepatocyte-specific NDRG3 knockout mice show hypoglycemia (Fig. 2A) lower intrahepatic glucose levels (Fig. 2C) but no changes in fasting hepatic glycogen content (Fig. 4A) or glycogen phosphorylase expression (Fig. 4E). These findings suggest that hypoglycemia at this stage is not related to impaired glycogen phosphorylase activity. In order to asses whether impaired gluconeogenesis could be explained lower blood glucose levels at 2 months of age, inclusion of data on gluconeogenic enzyme expression/activity affected and hepatic glucose-1 and 6-phosphate contents would be key.
-As hepatocyte-specific NDRG3 knockout mice share some features with hepatic GSD type 1 (i.e. hepatomegaly, hypoglycemia, hepatic glucose-/6-phosphate/glycogen accumulation) it would be relevant to evaluate hepatic triglyceride accumulation and lipogenic enzyme expression, which are increased in hepatic GSD type 1, as well.
-It is shown that the changes in hepatic glucose/glycogen metabolism in hepatocyte-specific NDRG3 knockout mice were paralleled by reduced GNMT expression and perturbed one carbon metabolism. Based on a previous study in GMNT knockout ice, the authors propose that these features are causally related (Fig. 7D). However, data that solidify a causal relationship between GNMT and Slc37a4/Pygl in hepatocyte-specific NDRG3 knockout mice are lacking. In order to support the claim of causality, such data should be included (e.g. a GMNT rescue experiment in hepatocyte-specific NDRG3 knockout mice) or the statements on causality should be rephrased/toned down.
-A key question that remains is how NDRG3 regulates the expression of enzymes involved glucose/glycogen/one carbon metabolism. Could the authors speculate on this, e.g. does NDRG3 act as a transcriptional regulator of specific genes involved in these pathways?
Round 2
Reviewer 1 Report
I thank the authors for addressing the majority of the previous comments. However, I still not agree with the conclusion that targeting NDRG3 may be used an anti-tumor strategy.
Considering the phenotype of the liver NDRG3 mice, the sentence in lines 729 to 731 should be removed.
Reviewer 2 Report
see the attached file

Reviewer 3 Report
The quality of this manuscript has significantly improved after these revisions. The authors have adequately addressed most of my comments and I would like to thank them for sharing their thoughts on the regulatory role of NDRG3 (point 5).
One minor comment remains:
Point 1: regard to the glucose production assays in primary hepatocytes (Fig 5A), these are expressed as 'relative glucose production'. It should be specified in the methods section/figure legend how relative production rates were determined.
I could not review the following data that were added by the authors:
Point 1: the PTT data which are referred to are not visible for this reviewer
Point 5: the AML12 data which are referred to are not visible for this reviewer
